# Exploring Spatial Patterns of Interurban Passenger Flows Using Dual Gravity Models

**DOI:** 10.3390/e24121792

**Published:** 2022-12-08

**Authors:** Zihan Wang, Yanguang Chen

**Affiliations:** Department of Geography, College of Urban and Environmental Sciences, Peking University, Beijing 100871, China

**Keywords:** dual gravity model, fractal, interurban passenger flows, Tencent location big data

## Abstract

Geographical gravity models can be employed to quantitatively describe and predict spatial flows, including migration flows, passenger flows, daily commuting flows, etc. However, how to model spatial flows and reveal the structure of urban traffic networks in the case of missing partial data is still a problem to be solved. This paper is devoted to characterizing the interurban passenger flows in the Beijing–Tianjin–Hebei region of China using dual gravity models and Tencent location big data. The method of parameter estimation is the least squares regression. The main results are as follows. First, both the railway and highway passenger flows can be effectively described by dual gravity models. A small part of missing spatial data can be compensated for by predicted values. Second, the fractal properties of traffic flows can be revealed. The railway passenger flows follow the gravity scaling law better than the highway passenger flows. Third, the prediction residuals indicate the changing trend of interurban connections in the study area in recent years. The center of gravity of the spatial dynamics has shifted from the Beijing–Tianjin–Tangshan triangle to the Beijing–Baoding–Shijiazhuang axis. A conclusion can be reached that the dual gravity model is an effective tool for analyzing spatial structures and dynamics of traffic networks and flows. Moreover, the model provides a new approach to estimating the fractal dimensions of traffic networks and spatial flow patterns.

## 1. Introduction

In research on urban and regional development, the prediction and analysis of spatial flows are very important. Spatial flows are generated by the interaction between georeferenced places through human migration, interurban traveling, daily commuting, and global trade [1,2]. According to the idea of a new science of cities advocated by Batty [3], cities should be treated not only as places in space but also as systems of networks and flows. To understand space, we must understand flows. If a spatial dataset for flows is complete for an urban system, we can adopt spatial interaction models based on entropy maximization for trip distribution and transportation system optimization [4,5]. However, Wilson’s models are not available in cases when key data is missing. In such instances, dual gravity models promoted by Chen [6] can be employed to describe and predict spatial flows between cities. Classical geographical gravity models are derived by analogy from Newton’s law of universal gravitation. Just as the gravitational forces of a pair of objects are equivalent to each other, the spatial flows between a pair of cities in this model are equivalent as well [7,8]. The dual gravity models, on the other hand, allow for different spatial flows between a pair of cities, which is more realistic. This pair of models can be derived from Wilson’s spatial interaction models and can be associated with fractal concepts [9]. Up to now, the main contribution of the models has been at the theoretical level, and this study will further show the value of the models at the practical level. This study uses dual gravity models to describe and predict interurban passenger flows in a specific metropolitan area in a case of missing partial data, and uses the case study to illustrate how to utilize the model in transport planning and regional development policy.

Data and models are both necessary for our understanding of urban and traffic networks. The success of science development rests on great emphasis on the role of quantifiable data and their interplay with models in scientific research [10]. The railway and highway passenger big data used in this article come from Tencent location big data, which can accurately reflect the trend of intercity population immigration [11,12,13,14]. The dual gravity models are employed to solve the following problems: characterizing the spatial flows of railway and highway passengers in our study area, estimating the fractal dimension of traffic networks based on flows, predicting the flow quantities of passengers, making up the missing data, and analyzing the structural and dynamic features of spatial flows. By fitting the dual gravity models with Tencent location big data, we model the spatial pattern of passenger flows. Using the residuals and outliers predicted by the models, we compare the gaps between the expectation and reality and discover an evolving trend in regional spatial patterns. The aim is to develop the methodology of geospatial analysis of spatial flows in cases of incomplete datasets. The Beijing–Tianjin–Hebei region, the study area of this research, is one of the most important metropolitan areas in China. Our methodology can be generalized to any other metropolitan area as well. The remainder of this paper is organized as follows: In Section 2, the model, study area, and foundational data are elucidated. In Section 3, the results and analytic process are presented. In Section 4 and Section 5, several questions are discussed, and the discussion is concluded by summarizing the main points of this study.

## 2. Methods and Data

### 2.1. Dual Gravity Models

Geographical gravity models are highly consistent with Tobler’s first law of geography, which is that ‘everything is related to everything else, but near things are more related than distant things’ [15]. The models’ status in urban geography is self-evident [16]. As early as 1713, Berkeley introduced the concept of gravity models in geography. Carey further developed the concept of gravity and potential models [7]. Ravenstein proposed the law of migration, which is proportional to the size of the settlements and inversely proportional to the square of the distance [17]. Since then, the law of gravity has been introduced in the study of human geography. Subsequently, many geographers have developed different forms of geographical models, such as Reilly’s law of retail gravitation [18], Stouffer’s intervening opportunity model [19], Stewart’s potential model [20], and Converse’s breaking-point model [21]. In the 1950s–1970s, many classical studies used geographical gravity models for human migration, urban travel, and trade areas analysis [8,22,23,24,25]. However, there are still some pending questions. One obvious one among them is the difference between geographical flows and physical gravity. By analogy to Newton’s gravity law, geographical gravity models are mostly expressed in the following terms:(1)$$I_{ij}=K\frac{P_{i}P_{j}}{r_{ij}^{b}},$$
where *I_ij_* denotes gravity between city *i* and *j*, *P_i_* denotes the size of city *i* (population is used to represent the size in this study), *P_j_* denotes the size of city *j*, *r_ij_* denotes the distance between city *i* and city *j*. Geographical gravity cannot be measured directly as physical gravity, so it always replaced by flows between cities. That means *I_ij_* and *I_ji_* have the same values, while in the real world they’re different. This problem can be solved by Chen’s dual gravity models.

Chen [6] proposed an improved global gravity model with a symmetrical expression, also known as the dual gravity model. The innovations of the model are as follows. First, traditional geographical gravity models, which are local, describe the relationship between one point and other points. For example, Mackay [26] used the gravity model to fit phone data for describing the relationship between Montreal and other surrounding cities. The dual gravity model can be used to describe the relationship between any pair of cities in a region and analyze global spatial interaction. Second, the dual gravity model in this study strictly distinguishes between the concepts of flow and gravity. Though the same equations were given in Chen’s previous studies, they used flows to represent gravity or used other data instead of flows to estimate parameters [6,9]. In the model used by us, cities exchange flows with different amounts, but the gravity between them is the same. Gravity is derived from the product of flows. In other words, after switching the origin and the destination, the gravity remains the same, while the flow always changes. In this way, the flows in the region constitute a directed network that can better characterize the spatial pattern of the region. The equations for dual gravity models are as follows:(2)$$T_{ij}=KP_{i}^{u}P_{j}^{v}r_{ij}^{-\sigma },$$
(3)$$T_{ji}=KP_{i}^{v}P_{j}^{u}r_{ji}^{-\sigma },$$
where *T_ij_* denotes the intensity of the flow from city *i* to city *j*, *T_ji_* denotes the intensity of the flow from city *j* to city *i*, *r_ij_* and *r_ji_* denote the distance between city *i* and city *j*. As for the parameters, *K* is a constant, *u*, *v*, and *σ* are gravity exponents, representing a set of cross-scaling exponents. From Equations (2) and (3), a standard gravity model can be derived as follows:(4)$$I_{ij}={\left(T_{ij}T_{ji}\right)}^{1/(u+v)}=K^{2/(u+v)}\frac{P_{i}P_{j}}{r_{ij}^{2\sigma /(u+v)}}=G\frac{P_{i}P_{j}}{r_{ij}^{b}}.$$
The expressions of the parameters in Equation (4) follow:(5)$$G=K^{2/(u+v)},$$
(6)$$b=\frac{2\sigma }{u+v},$$
where *G* denotes the gravity coefficient, and *b* denotes the distance decay exponent, which proved to be the average fractal dimension of the corresponding urban size measurement [6,9]. In this sense, Equations (2)–(4) are in fact fractal gravity models. The flows can reflect the size of the attractive forces between cities, but flows and attractive forces are not equal. The most intuitive performance of the differences between them is: *T_ij_* is always not equal to *T_ji_*, while *I_ij_* is equal to *I_ji_*. The asymmetry of the flows is embodied in the dual gravity model as the difference between the scaling exponents, *u*, and *v*. The essence is that the sizes of the origin city and the destination city have different influence weights on the size of flows. This is one of the novelties of the dual gravity models used in the study.

The theoretical foundation of the dual gravity models lies in the principle of entropy maximization. Based on the allometric scaling relation between city size and spatial flow quantity, Equations (2) and (3) can be derived from Wilson’s revised spatial interaction model. Wilson’s spatial interaction models can be derived from the postulate of entropy maximization [27,28,29]. This suggests that allometric scaling and entropy maximizing processes are basic rationales of dual gravity models. In fact, entropy maximization is a very important principle in geographical analysis [30,31,32,33,34,35,36].

The model parameter values can be estimated by least squares calculation. Converting Equation (2) into logarithmic form, we have
(7)$$\ln T_{ij}=\ln K+u\ln P_{i}+v\ln P_{j}-\sigma \ln r_{ij}.$$
In this way can we use ln*T_ij_* as the dependent variable, and ln*P_i_*, ln*P_j_*, and lnr*_ij_* as independent variables for multiple linear regression. After getting the model’s parameters by least squares calculation, we re-forecast the passenger flow matrix. Some information about the spatial pattern of regional migrations in the Beijing–Tianjin–Hebei Region is hidden in the residuals between the expected values and the real values.

### 2.2. Study Area and Data

The study area includes Beijing Municipality, Tianjin Municipality, and Hebei province. The network of cities in the Beijing–Tianjin–Hebei region constitutes one of the three major urban systems in mainland China (Figure 1). They have considerable influence on the economic development of north China and even the whole country. At present, the coordinated development of the Beijing–Tianjin–Hebei region has become a Chinese national strategic plan. In 2016, the *Beijing–Tianjin–Hebei National Economic and Social Development Plan during the 13th Five-Year Plan period* was issued. The plan put forward nine tasks for improving the overall development of that urban system. One of the nine tasks was to focus on accelerating the construction of integrated transportation construction, and ‘build the Beijing–Tianjin–Hebei on the track’. This study takes the Beijing–Tianjin–Hebei region as the research object, firstly, to explore its spatial pattern and propose targeted development suggestions, and secondly, to provide a paradigm that other metropolitan areas around the world can refer to for transportation planning evaluation.

The main data used in this article came from the Tencent location big data platform (https://heat.qq.com/ accessed on 1 July 2019). Tencent location big data, which can reflect destinations and scales of population flow between cities, is generated in real-time by the built-in location calls of Tencent applications such as WeChat, QQ, and Tencent Maps. Location software development kits (SDKs) are embedded in Tencent applications. A location SDK is a set of location-based services (LBS) positioning interfaces that can use a positioning application programming interface (API) to obtain positioning results, inverse geocoding, and geofencing. When users of those applications actively call the location SDKs, data will be generated on Tencent servers. The data are processed by specific modules such as distributed data warehouse, real-time computing, and data bank. Then, real-time big data with accurate location can be obtained. They can be used for personal travel planning, business logistics planning, municipal transportation planning, and disaster relief planning. At present, Tencent applications have about 800 million daily active users, which covers more than 70% of the total Chinese population. Location calls per day exceed 55 billion. Location check-in times per day exceed 60 million. Although we lack some details about Tencent LBS, such as sampling frequency and penetration rate, previous studies have shown that among similar big data, Tencent location big data is more accurate and reliable, with an overall mapping accuracy of 88.9% in China [11,37,38,39].

The passenger flows provided by Tencent location big data describe the intensity of daily intercity population migrations. The passenger flows are calculated comprehensively by passenger numbers, transportation modes, and travel distance. The flows can be divided into three categories according to transportation modes: air passenger flows, railway passenger flows, and highway passenger flows. We do not collect the air passenger flows, because the travel distance in the Beijing–Tianjin–Hebei region is limited. The actual air passenger flow values are always zero. The weight of travel distance cannot be excluded from flows, but the impact of the weight on analytical results is not significant because the travel distance in the region is very slight compared to nationwide. The population data used in this article are the urban annual average populations provided by *China City Statistical Year Book 2017* (See Appendix A for processed datasets).

### 2.3. Data Processing

After selecting transportation modes, we collected data on the 1st of each month from April 2018 to March 2019 and recorded it in several matrices. To avoid random disturbance, we averaged the corresponding rows and columns in the matrices to get an average highway passenger flow matrix and an average railway passenger flow matrix. During data collection, it was found that the dataset contains obvious outliers. The outliers were passenger flows during the Spring Festival, May Day, and Chinese National Day holidays because a large number of Chinese choose to travel during these holidays. This is a characteristic of China’s intercity mobility and is specially studied by many scholars. However, we focused on passengers’ daily travel patterns. These peak passenger flows are 50% higher than the normal flows. Data cleaning was therefore necessary for that dataset. The variation of the regression parameters after data cleaning is not significant from an overall perspective (Table 1). The sample sizes of railway passenger flows and highway passenger flows are 144 and 131, which are large enough to make some outliers not seriously affect the final fitting effect of models to data. Time-of-day and day-of-the-week variations did not exceed the outliers on holidays, and their effects can be ignored.

## 3. Results

### 3.1. The Spatial Pattern of Passenger Flows in the Study Area

In light of the average passenger flow matrix, the spatial pattern of the regional passenger flows can be drawn. The results are displayed in Figure 1. As can be seen, the passenger flows of Beijing city, Tianjin city, and Shijiazhuang city, which is the capital of Hebei province, are at the forefront of the Beijing–Tianjin–Hebei region, and these three cities are the cores of the regional population migrations. The passenger flows of Qinhuangdao city, Zhangjiakou city, and Chengde city are the lowest in the region. These three cities are not only located at the edge of the region but also represent the trough of regional population migrations. From an overall point of view, the northern and southern ends of the region are passenger flow valleys, and the middle section has more passenger flows. Beijing and its main external connection destinations which are southeast and southwest are the passenger flow peaks (Table 2).

During the study period, railway passenger flows were greater than highway passenger flows in every city. It shows that railways are the main mode of intercity migration in the Beijing–Tianjin–Hebei region. In other words, the people in that region prefer railways to highways for intercity traveling. According to *the Statistical Bulletin of the development of the Chinese Transportation Industry* issued by the Ministry of Transport of the People’s Republic of China, the railway passenger turnover exceeded the highway passenger turnover from 2014 to 2018, and its market share is still growing. The passenger turnover is the product of the number of passengers and the distance traveled. Railways have long been considered the focus of the future development of intercity passenger transportation in China.

Because Tencent location big data only provided the top 10 inflows and top 10 outflows for every city, the passenger flow matrix is not complete. The missing values are too small to be in the top 10. Marking the missing values on the map and using arrows indicating directions, we can generate Figure 2. It can be seen that most missing values are vertical and vertical intercity connections are weaker than horizontal intercity connections, mainly because the northern and southern areas are not closely connected. That is related to the fact that the vertical span of the region is greater than the horizontal span. The Euclidean distance between the northmost Chengde city and the southmost Handan city in the region is 565 km, while the Euclidean distance between the westmost Zhangjiakou city and the eastmost Qinhuangdao city is 410 km. The weaker connections between the north and south of the Beijing–Tianjin–Hebei region are the inevitable consequence of its larger spatial span. This complies with Tobler’s first law of geography.

There are more missing values of highway passenger flows than railway passenger flows, because the spatial scales of the two transportation modes are different. Highways have a smaller spatial scale and denser tracks than railways. For example, Xilin Gol League, which is located in the middle of Inner Mongolia of China, ranked 6th in the top 10 road passenger inflows of Chengde city on 12 April but ranked below the top 10 railway passenger inflows. Highways can connect Chengde city and Xilin Gol League directly, while railways cannot. There are more cities not belonging to the Beijing–Tianjin–Hebei region such as Xilin Gol League in the top 10 highway passenger flows than railway passenger flows, resulting in more missing values for the Beijing–Tianjin–Hebei region. There are some differences in the arrows of the missing values of highways and railways, but they are mainly vertical (See attached Appendix A).

### 3.2. Results of the Dual Gravity Modeling

The combination of model and data constitutes the foundation for our spatial analysis. On the one hand, data generate stylized facts and put constraints on models, on the other hand, models are essential for comprehending the processes at play and how the system works [10]. The model parameters can be estimated by means of least squares regression analysis based on log-linear relationships and the above processed data (See Appendix A for calculation processes). For railway passenger flows, the log-linear fitting results of the dual gravity model are as follows:(8)$$\ln{\hat{T}}_{ij}=-2.4228+0.2374\ln P_{i}+0.2176\ln P_{j}-0.4412\ln r_{ij}.$$
Converting Equation (8) into the general form as
(9)$${\hat{T}}_{ij}=0.0887P_{i}^{0.2374}P_{j}^{0.2176}r_{ij}^{-0.4412}.$$
The dual expression of Equation (9) is as follows:(10)$${\hat{T}}_{ji}=0.0887P_{i}^{0.2176}P_{j}^{0.2374}r_{ji}^{-0.4412}.$$
Using Equations (5) and (6), we can estimate the values of the gravity coefficient *G* and distance exponent *b*, and the results are as below:(11)G=0.00002373, b=1.9391.
Substituting the parameter values into Equation (4) yields a standardized gravity model, i.e., a pure gravity model based on railway passenger flows, as follows:(12)$$I_{ij}=0.00002373\frac{P_{i}P_{j}}{r_{ij}^{1.9391}}.$$
The value of *b* comes between 1 and 3, indicating a fractal structure of the railway passenger flow system. For highway passenger flows, the fitting results are as follows:(13)$$\ln{\hat{T}}_{ij}=3.3689+0.1390\ln P_{i}+0.1195\ln P_{j}-1.0743\ln r_{ij}.$$
Transforming Equation (13) into the general form as
(14)$${\hat{T}}_{ij}=29.0477P_{i}^{0.1390}P_{j}^{0.1195}r_{ij}^{-1.0743}.$$
Transforming Equation (13) into the general form as
(15)$${\hat{T}}_{ji}=29.0477P_{i}^{0.1195}P_{j}^{0.1390}r_{ji}^{-1.0743}.$$
The gravity coefficient *G* and distance exponent *b* can be approximately evaluated as below:(16)G=210010627962.3300, b=8.3135.
Inserting the parameter values into Equation (4) yields a pure gravity model based on highway passenger flows, as below:(17)$$I_{ij}=210010627962.3300\frac{P_{i}P_{j}}{r_{ij}^{8.3135}}.$$
The value of *b* departs from the reasonable range, 1 and 3, suggesting a broken fractal structure of the highway passenger flows system.

As can be seen, the dual gravity models can be well fitted to the spatial flow data of interurban passengers in our study area. Both the railway and highway passenger flows can be described with the fractal gravity models. The main test statistics of the railway model and the highway model are summarized in Table 3. In a word, the two models can pass the tests with a 99% confidence level. However, the parameter meanings are different. Comparing the two models, we can find the following characteristics. First, the positivity and negativity of parameters are reasonable. Population sizes have positive effects on intercity flows and distances have negative effects on intercity flows. Next, the absolute value of the coefficient of ln*r_ij_* in the railway model is close to twice that of ln*P*, while in the highway model, the ratio is close to eight times. This shows that distance decay effects in highway transportation are more significant than that in railway transportation.

The distance decay exponent of gravity models bears the property of a fractal parameter. The *b* value is proved to equal the product of the Zipf exponent and fractal dimension of central place networks [40], that is
(18)$$b=qD_{f}=D_{P}\to 2,\;$$
where *q* refers to the Zipf exponent of city size distributions [41,42], *D_f_* denotes the fractal dimension of central place networks (see Christaller [43] for central place theory), and *D_p_* is the average dimension of the urban population [6]. The fractal dimension of central place networks is actually the similarity dimension of the urban hierarchy [9]. In theory, the Zipf exponent approaches 1, and the similarity dimension of central place networks changes around 2. Therefore, the distance exponent is about 2. The distance exponent based on the railway passenger flows is about *b* = 1.9393, which is close to 2. By contrast, the distance exponent value based on the highway passenger flows is around *b* = 8.3118, which departs significantly from 2. This suggests that the action at a distance of highway passenger flows is significantly weaker than that of railway passenger flows. The fractal patterns of passenger flows may be determined by their supportive system, including population distribution and transportation network. The spatial distribution of the population proves to have fractal properties [44]. More empirical analysis shows that the traffic network has a fractal structure, and fractal studies of road systems involve interurban traffic networks [45,46], intra-urban roads and streets [47,48,49,50,51], and suburban and exurban areas [52,53]. Population patterns influence Zipf’s distribution of city sizes, and traffic patterns influence networks of cities. According to Equation (18), the fractal structure and parameters of spatial passenger flows depend on both the city rank-size distribution and the network of cities. The mechanism is not clear for the time being and remains to be explored deeply in future studies.

The dual gravity model can be used to predict passenger flows in our study area, so as to make up for the missing data. In fact, all mathematical modeling has two major aims: explanation and prediction [54,55,56]. Using Equations (9) and (10), we can predict the railway passenger flows, and using Equations (14) and (15), we can predict the highway passenger flows. Although the data in Table 2 are incomplete, we can obtain the complete Table 4 by making predictions for the missing values. The missing values cannot be obtained from other sources. Using a mathematical model to generate missing data is actually to give out-of-sample predicted values. The effect of generating out-of-sample predictive values of a model depends on the effect of generating in-sample predictive values. In linear regressive analysis, the quality of in-sample predicted values can be evaluated by the standard error of regression [57]. To avoid the influence of variables dimension, the standard error is always converted into a coefficient of variation (CV). If the CV value is less than 0.1, the accuracy of the in-sample prediction of a model is very good; if the CV value is greater than 0.15, the prediction error is large (See Appendix B for how to calculate CV). These are empirical criteria, and there is no absolute benchmark. As far as the railway passenger flow models are concerned, the CV values are between 0.1 and 0.15; for the highway passenger flow models, the CV values are greater than 0.15 (Table 3). A conclusion can be reached that the prediction effect of the railway passenger flow gravity model is acceptable, while that of the highway passenger flow prediction model is not too satisfactory. If the predicted value of the logarithmic linear expression of a gravity model is satisfactory, the predicted value of the original nonlinear gravity model will also be acceptable. By the way, since the statistical test theory of linear models is the most mature, all nonlinear models that can be linearized are best evaluated with the help of linear transformation results. On the whole, the predicted values are close to the observed values (Table 4). The difference values between Table 2 and Table 4 are the prediction residuals. The residuals can be used to analyze the characteristics and changing trends of the traffic pattern in the Beijing–Tianjin–Hebei region. What is more, the pure gravity models, Equations (12) and (17), can be employed to calculate the spatial connection strengths based on the railway and highway passenger flows between any two cities in the study area (See Appendix A).

By contrast, the standard gravity model can be employed to calculate the attractive force between cities. The attractive force reflects spatial connection strength and can be treated as a measure of spatial strength of association among cities. Flow is a spatial vector with direction, which is divided into outflows (*T_ij_*) and inflows (*T_ji_*). Therefore, the spatial flows are generally asymmetric quantities (*T_ij_* ≠ *T_ji_*). The connection strength is a spatial scalar without direction, so it has symmetry (*I_ij_* = *I_ji_*). With the help of Equation (4) and the model parameters displayed in Table 1, as well as the dataset in the attached files, it is easy to calculate the connection strength based on given spatial flows between any two cities. For example, using Equation (12), we can compute the spatial association strength of the cities in the study area based on railway passenger flows (Table 5). Summating the data in the table by row or column gives the total connection strength of each city (Figure 3). The connection strength can be regarded as a standardized gravitational measure. Analyzing association strength can provide a reference for traffic network planning or urban attractiveness research.

Now, let us examine the spatial connection strength patterns of the cities in the Beijing–Tianjin–Hebei region. Comparing the attractive forces measured by railway passenger flows and predicted by Equation (12) with the actual attractive forces, we can find the differences between theoretical expectations and reality. The calculated results are presented in Figure 4. As for the total connection strength of each city, see Figure 3. The actual network of spatial connection strength based on railway passenger flow in our study area takes Beijing city as the first-level single core, and Tianjin city and Shijiazhuang city as the secondary cores. The main axis of the actual network is the southeast and the southwest starting from Beijing city. The predicted patterns of spatial connection strength based on railway passengers take Beijing city and Tianjin city as the first-level dual cores, and the entire network is distributed radially from the center, Beijing city, to the surroundings. The cities whose actual total connection strength is significantly greater than the predicted values included Beijing city, Shijiazhuang city, Baoding city, and Langfang city. The former three cities happen to form the southwest axis starting from Beijing, named ‘the Beijing–Baoding–Shijiazhuang development axis.’ The cities whose actual flows are significantly lower than the predicted values included Tianjin city and Tangshan city. These cities all lie in the northeast of the region. *The outline of the plan for Coordinated Development of the Beijing–Tianjin–Hebei region* issued in 2015 clearly stated a development framework consisting of ‘one core, two main cities, three axes, four districts, and multiple nodes.’ Among them, ‘one core’ refers to the capital, Beijing city. ‘Two main cities’ refers to Beijing city and Tianjin city, and the Beijing–Tianjin linkage is expected to be the development engine of the region. ‘Three axes’ refers to the Beijing–Tianjin axis, the Beijing–Baoding–Shijiazhuang axis, and the Beijing–Tangshan–Qinhuangdao axis. These three axes will serve as industrial regions and metropolitan areas. It can be seen from Figure 4c that the ‘one core’, Beijing city, is more important in the region than predicted. Tianjin city as one of the ‘two main cities’ has a close connection with Beijing, but its connections with other cities are weaker than predicted, especially the connection with Tangshan city. Among the ‘three axes’, the actual connection strength of the Beijing–Tianjin axis is close to the predicted value. The actual connection intensity of the Beijing–Baoding–Shijiazhuang axis is significantly stronger than predicted. The actual connection intensity of the Beijing–Tangshan–Qinhuangdao axis is lower than predicted, and the attractive force between Beijing city and Tangshan city is significantly lower than predicted. However, for a long time, Beijing–Tianjin–Tangshan was the important and energetic urban triangle of the Bohai Bay Economic Zone. Nowadays, the passenger flow in this triangle area is not active enough. What is the problem? This is worthy of further investigation and study.

## 4. Discussion

The results of data analysis show that dual gravity models are available to describe the spatial pattern of railway and highway passenger flows in the Beijing–Tianjin–Hebei region. The goodness of models fitted to observational data is satisfying. The predicted values of models roughly match the actual values. A series of spatial analyses can be conducted by using the models. First, the models can be used to predict the passenger flows and complement the small part of the missing data approximately. Second, the fractal structure and abnormal fractal structure of the spatial flows can be revealed. Third, the differences between predicted values and actual values of passenger flows can be compared using the models. The future evolutionary trends of transport patterns in the Beijing–Tianjin–Hebei region can be judged by the residuals of the prediction.

However, if the gravity models based on flows are transferred to the standard gravity model, it is found that the railway passenger flow model differs significantly from the highway passenger flow model where parameter values are concerned. The standardized distance exponent of the railway gravity model is close to 2, which is not far from the predicted value in theory [6,9]. However, the standardized distance exponent of the highway gravity model is more than 8, which is more than four times the railway distance exponent. That means the railway passenger flows in the Beijing–Tianjin–Hebei area decay normally with a normal fractal structure, while the highway passenger flows quickly decay and deviate from the normal fractal structure. Fractal suggests a kind of optimized structure in nature [58]. A fractal object can occupy its space in the best way. The departure of an urban phenomenon or a traffic network from a fractal structure suggests a kind of evolution fluctuation or development obstacle. The railway passenger flows show the long-distance effect, and the highway passenger flows show the localization property to some extent. People choose highway transport generally for short trips. For long trips, they mostly choose railway transport. It can be seen that railway passenger flows follow the first law in geography better than highway passenger flows.

The reasons for the outliers should be discussed. In Section 3.2, the spatial pattern of the prediction bias is depicted. The residuals that are not within plus or minus two standard deviations will be discussed next. There is only one outlier in the highway passenger flow model. The standard deviation of highway passenger flow from Beijing city to Baoding city is 2.04. The railway passenger flow model has more outliers than the highway model (Figure 5c). The spatial pattern of railway outliers is highly related to the unbalanced distribution of the regional intercity railway network (Figure 5a). Outliers that are beyond two standard deviations appear in the middle of the region, where there are densely distributed railways. Outliers that are below two standard deviations appear in the north of the region, where there are mountains and sparsely distributed railways. By contrast, the distribution of the regional highway network is more even (Figure 5b), resulting in fewer outliers in the highway model.

The flows between cities are related to regional gravity and spatial interaction. The common models for predicting population flows are Wilson’s spatial interaction models [28]. However, Wilson’s models are not applicable to this study and cannot be compared with the dual gravity models in our case study. First, one of the prerequisites of Wilson’s models is data integrity. It is desirable to have complete population flows between any two cities. Taking a step back, we have to at least get the total inflows and outflows of every city to build the models. However, we mined most of the flow data between cities and obtained neither exact total inflows nor outflows for every city nor the total population flows of all cities. Therefore, Wilson’s models (total flow constrained model, singly constrained model, doubly constrained model) could not be used sufficiently in our study. Second, Wilson’s models are essentially nonlinear programming models to optimize traffic flows. The models are very useful for urban planning and regional transportation planning. In a word, Wilson’s spatial interaction models apply to normative research aimed at urban and transportation planning and the gravity model applies to behavioral research oriented to empirical analysis. When transportation flow data are incomplete, dual gravity models can predict flows in urban space indirectly using urban attractive forces and avoid using the total population flow data of every city.

As the first application of the dual gravity models to interurban passenger flows, this study further standardizes related concepts and clearly distinguishes the difference between flow and gravity. The reason that flow is asymmetrical and gravity is symmetrical is that the sizes of the flow origin cities and destination cities have different scaling exponents. Specifically, the population flow from large cities to small cities in the Beijing–Tianjin–Hebei region tends to be larger than that from small cities to large cities. According to city statistical yearbooks between 2016 and 2018, the resident populations of large cities in the region, such as Beijing city and Tianjin city, have been declining year by year, which is consistent with the trend reflected in this study.

Some new gravity models with satisfactory data fitting and predictive abilities have been developed in recent years, such as the Poisson pseudo-maximum likelihood estimation gravity model, the eigenvector spatial filtering hurdle gravity model, and the network spatially filtered Poisson migration model [59,60,61]. These models consider the spatial autocorrelation problem and introduce some control variables. In practice, these models can be used for urban governance because they can tell which determinants affect migrations more. Our models can be used for macro-regional planning because the changing regional development focus trend can be detected, and the transportation network fractal structure can be characterized with our models. Compared with these newly developed models, our models have both advantages and deficiencies. The advantages of our study lie in three aspects. First, in the case of missing a very small part of spatial data, that is, the spatial dataset is not complete, we can make up the grid data and reveal the spatial distribution characteristics of passenger flows in the study area using dual gravity models. Second, ideas from fractal theory are employed to reveal the deep structure of the traffic network. Third, the prediction residuals are used to analyze the evolutionary trends of the spatial pattern of urban systems in the study area. For any other metropolitan area, the dual gravity models can be used to explore its spatial pattern as well. The steps include: obtaining the interurban flow matrix (complete or incomplete), estimating the model parameters by log-linear fitting, comparing the predicted values with the actual values, and finally refining the analysis with reality.

The shortcomings of the study are in three areas. First, the urban population size data and population flow data for analysis are not an exact match temporally. This defect has no significant effects on population flow analysis because of the following two aspects. On one hand, the urban population size is relatively stable. Our research relies heavily on the relative sizes between cities instead of absolute sizes. The relative sizes of cities in the Beijing–Tianjin–Hebei region will not change much in a few years. On the other hand, urban sizes have a kind of time-lag effect on population flows. If the interaction between urban sizes is considered as input and the population flows are considered as output, there is a time lag between sizes (inputs) and flows (outputs). This is known as the response delayed effect reflected on spatial interaction in geographic space [62]. Second, the local gravity analysis is not carried out. The dual gravity models used in our study are global models to describe urban attractive forces based on traffic data. The models can characterize the traffic flow pattern in a region from a holistic perspective. However, if researchers want to reveal the spatial–temporal evolutionary characteristics of population flows in the study area systematically and thoroughly, it is necessary to use local gravity models to explore the spatial heterogeneity of urban population migration behavior. What is more, the dual gravity models are a kind of fractal gravity models in essence. The fractal characteristics of population migration in the Beijing–Tianjin–Hebei region require further systematic exploration. Third, the influencing factors of traffic flow only include the urban population size and interurban distance. Other control variables such as industrial scale and infrastructure level have not been taken into account. There are two central variables in the study of the spatial dynamics of urban development: one is population, and the other is wealth [63]. Population represents the first dynamic of city development [64]. Therefore, if a variable is adopted to study cities, population is the first choice; if two variables are employed to research cities, population and wealth are the best two choices. In this paper, population and distance serving as control variables were employed to measure urban gravity. On the other hand, there are allometric scaling relationships between urban population and other urban variables such as industry structure and amenities [65]. Based on the allometric scaling relation, other variables can be used as alternative variables of urban population for gravity analysis. In future research, if the necessary datasets are obtained, industry and infrastructure will be used as the control variables to replace the population sizes of cities to conduct a dual gravity analysis.

## 5. Conclusions

The dual gravity model fits the railway and highway passenger flows well with a few prediction outliers in the Beijing–Tianjin–Hebei region. The highway model has almost no outliers because the distribution of highways in the study area is relatively even. The spatial pattern of the railway model outliers is highly consistent with the spatial distribution of the regional railway network. The research goal of this paper was to reveal the real problems by using dual gravity models, and our analytical process can be generalized to any other metropolitan area. Based on the above results and discussion, the main conclusions could be drawn as follows.

First, dual gravity models can effectively describe and predict the spatial pattern of passenger flows between cities. In terms of railway and highway passenger flows in the Beijing–Tianjin–Hebei region, observation values and prediction values have a satisfying overall match, and some local differences between them can reflect gaps between reality and expectation. Partial missing data can be made up by prediction values. Then, spatial interaction analysis based on entropy maximation can be conducted using this complete dataset to provide planning guides for future transportation development in the study area. There is a key problem that can be illustrated by gravity modeling effects of the regional passenger flow distribution in the study area: the distance decay effect still dominates the changing trends of spatial patterns in human geographic systems. Due to rapid changes in new technologies, many scholars doubt whether the geographic distance decay law is still valid. The modeling effects in our research gave a positive answer to this question. The distance exponent in the railway model is very close to the theoretical expectation, which indicates that the role of railways in the transportation system of mainland China has the characteristics of large scale and stability. By contrast, the distance exponent in the highway model is more than four times the theoretical expectation, which indicates both the short-rangeness and the poorer stability of railway passenger flows.

Second, dual gravity models reflect a fractal pattern and variation characteristics of passenger flow distribution in the study area. As mentioned earlier, dual gravity models are essentially fractal models and normalized distance exponents are fractal parameters. The parameters reflect the spatial pattern of urban population distribution, linking both Zipf’s rank-size distribution and central place networks. Mathematical models reflect macro patterns and model parameters reflect the behaviors of microelements. In terms of the Beijing–Tianjin–Hebei region, the railway passenger flows obey a gravity model in which distance decays based on a power law. The normalized distance exponent in the railway model is between 1 and 2 and is a fractal dimension corresponding to normal rank-size distribution and central place networks. It can be seen that railway passenger flows in the region are consistent with fractal patterns at both macro and micro levels. The highway passenger flows also obey a gravity model with distance decay based on a power law and that shows a macrostructure consistent with fractal patterns. However, the parameter reflecting micro-level element interactions in the highway model deviates from a reasonable range. As a fractal parameter, the distance exponent in the highway model should be between 0 and 3. In fact, that distance exponent is greater than 8 which is ridiculously high. The high distance exponent seems to imply a high distance decay effect. However, it exceeds the reasonable theoretical range and shows a deviation of railway passenger flows from a fractal structure. The reason for that phenomenon needs to be further explored.

Third, the global consistency and local differences between the predicted values based on gravity models and the observed values of passenger flows can reflect changing trends of spatial patterns in the urban system. In terms of the Beijing–Tianjin–Hebei region, the past spatial pattern with the Beijing–Tianjin–Tangshan triangle as the center is changing to a new pattern with the Beijing–Shijiazhuang–Tangshan large triangle as the center. That large triangle concludes Tianjin city and Baoding city. From the perspective of observed values and predicted values, the passenger flows in Beijing city, Tianjin city, and Tangshan city remain absolutely prominent throughout the study area. However, if changing trends are considered, passenger flow connections between Tianjin city, Tangshan city, and other cities show signs of relatively decreasing while passenger flows between Beijing city, Shijiazhuang city, Baoding city, and other cities are increasing. The prediction residuals of Tianjin city and Tangshan city with other cities are mostly negative. The sum of prediction residuals for these two cities is significantly negative. By contrast, the prediction residuals of passenger inflows and outflows in Beijing city, Shijiazhuang city, and Baoding city are mostly positive. The sum of prediction residuals for these three cities is significantly positive. It means that the external connections of Tianjin city and Tangshan city are lower than expected while the external connections of Beijing city, Shijiazhuang city, and Baoding city are higher than expected. Though the population size of Beijing city is smaller than that of Shanghai city, Beijing city is the primate city in terms of politics in mainland China and its function speaks for itself. The passenger flow connections of the Beijing–Baoding–Shijiazhuang axis, which runs northeast–southwest, are gradually strengthening. This may be influenced by China’s policy of developing the Beijing–Tianjin–Hebei region and is also related to the impact of the Beijing–Guangzhou railway. Tianjin city is the gateway for Beijing city to open up to the outside and the development pattern of the Beijing–Tianjin–Tangshan triangle is a product of China’s reform and opening up. The overall pattern in the region seems to be changing from outward-oriented development to inward-oriented development.

## Figures and Tables

**Figure 1 entropy-24-01792-f001:**
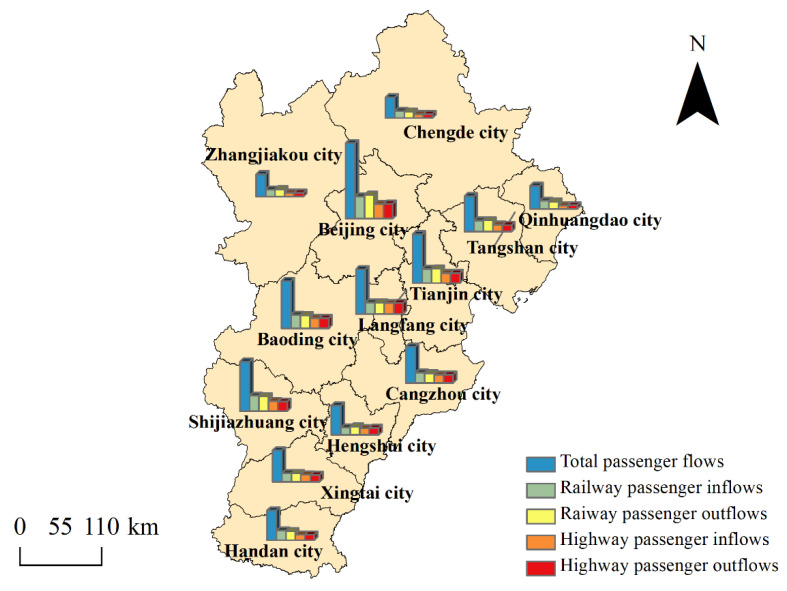
The spatial pattern of passenger flows in the Beijing–Tianjin–Hebei region. Total passenger flows are the sum of railway passenger flows and highway passenger flows.

**Figure 2 entropy-24-01792-f002:**
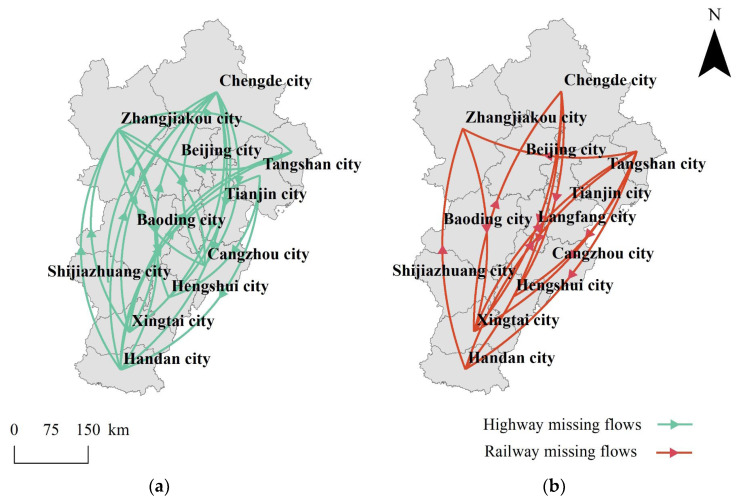
Missing values for the Beijing–Tianjin–Hebei region. The directions of the arrows are the same as the passenger flow directions. (**a**) Highway missing values; (**b**) railway missing values.

**Figure 3 entropy-24-01792-f003:**
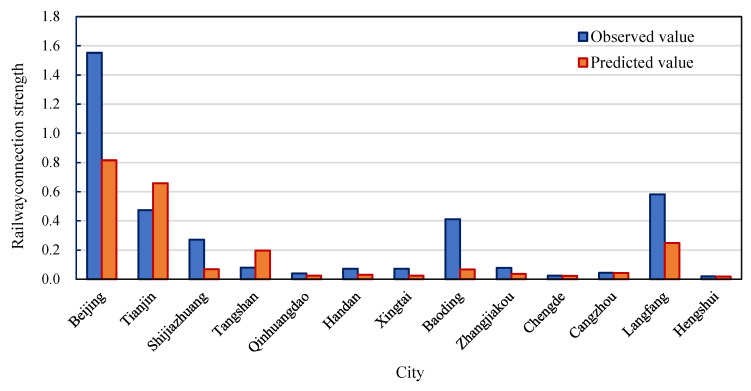
Comparison of observed values and predicted values of spatial connection strength based on railway passenger flows. Due to partial missing observed data points, the actual connection strength is not completely accurate.

**Figure 4 entropy-24-01792-f004:**
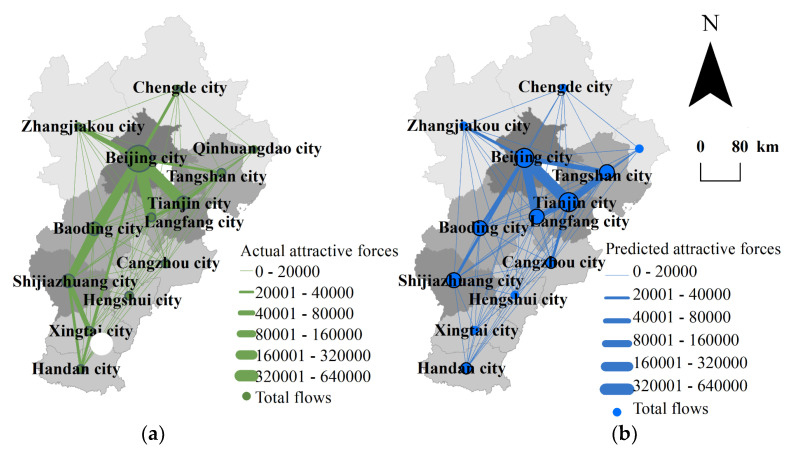
The actual values, predicted values, and residuals of regional railway passenger forces. (**a**) Actual attractive forces calculated based on actual flows *T_ij_*, *T_ji_*, and Equation (4); (**b**) predicted attractive forces estimated based on population *P_i_*, *P_j_*, and Equation (12); (**c**) residuals between actual values and predicted values. The little blank area beside Beijing city is Xianghe county, which is an enclave of Langfang city, and its data is missing.

**Figure 5 entropy-24-01792-f005:**
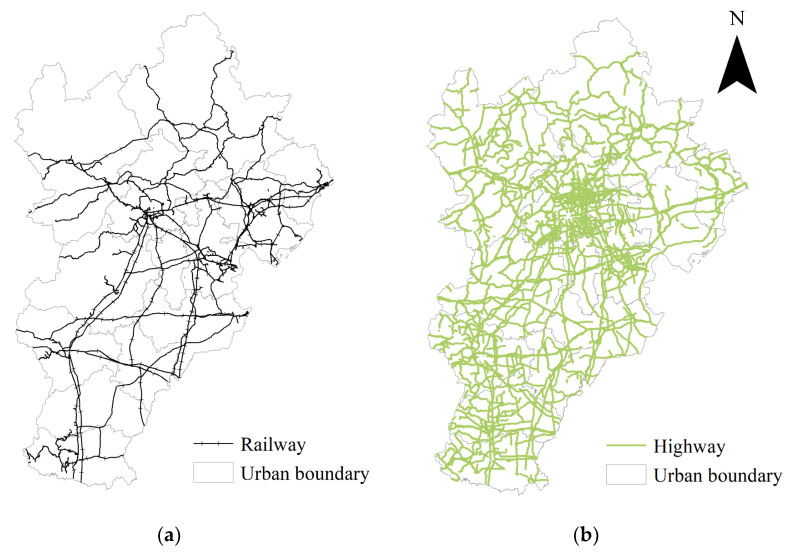
The railways, highways, and railway passenger flow outliers in the Beijing–Tianjin–Hebei region. (**a**) Railways; (**b**) highways; (**c**) railway outliers beyond two standard deviations. Highway and railway vectors are from 2018 data open-sourced from the OpenStreetMap platform.

**Table 1 entropy-24-01792-t001:** The parameter values of dual gravity models and standard gravity models before and after data cleaning.

Model	Parameter	Railway Model		Highway Model	
		Before Data Cleaning	After Data Cleaning	Before Data Cleaning	After Data Cleaning
Equation (2)	*K*	0.0974	0.0887	26.5124	29.0477
	*u*	0.2356	0.2374	0.1347	0.1390
	*v*	0.2161	0.2176	0.1214	0.1195
	*σ*	0.4335	0.4412	1.0330	1.0743
Equation (4)	*G*	0.0000	0.0000	131,036,533,086.4270	210,010,627,962.3300
	*b*	1.9194	1.9391	8.0680	8.3135

Note: (1) Data Resources: Railway passenger flows provided by Tencent location big data. (2) The model parameters of Equation (3) can be obtained by exchanging the values of parameters *u* and *v*.

**Table 2 entropy-24-01792-t002:** Partial processed results for railway passenger flows of the Beijing–Tianjin–Hebei region (average values).

	Beijing	Tianjin	Shijiazhuang	Tangshan	Qinhuangdao	Handan	Xingtai	Baoding	Zhangjiakou	Chengde	Cangzhou	Langfang	Hengshui
Beijing	0	19.0700	14.3900	10.5200	9.7900	11.2500	9.4300	18.5000	13.2800	10.0800	9.3200	21.1100	8.1222
Tianjin	18.1900	0	7.3100	10.5700	6.7800	5.3500	4.7000	7.5200	4.3000	3.8600	9.4100	9.7900	4.8222
Shijiazhuang	13.0200	7.1667	0	5.9700	4.8444	9.7500	11.3300	13.0800	4.5300	3.7300	6.2600	5.9000	7.3600
Tangshan	9.8900	10.4400	6.2400	0	8.6200	5.6000	4.3500	4.9300	3.0800	4.7500	4.2000	5.0500	2.9333
Qinhuangdao	9.0500	6.6600	5.0500	8.5800	0	na	na	4.9167	na	3.5556	4.3000	4.1400	na
Handan	9.9700	4.7800	9.7700	4.5000	na	0	8.4800	5.4600	na	3.4000	4.9000	3.8667	3.3000
Xingtai	8.1200	4.1400	11.1900	3.5000	na	8.4500	0	5.3800	3.3000	na	3.2750	4.0000	3.8500
Baoding	17.0100	7.3700	13.2700	4.8900	4.1333	5.4800	5.3500	0	4.3800	3.1900	5.3800	6.4200	3.8111
Zhangjiakou	12.4700	4.5400	4.4500	3.3700	3.4000	3.8000	na	4.4300	0	2.5000	3.9500	3.3300	3.3000
Chengde	9.4600	3.7500	4.0100	4.6900	3.1700	na	na	3.3800	3.0000	0	3.8000	3.0400	na
Cangzhou	8.6100	8.8700	6.7900	4.2900	4.0333	4.8000	3.4111	5.5700	2.6000	2.3750	0	4.7800	6.0600
Langfang	19.3800	9.6700	6.1200	5.0400	4.1200	3.7000	3.7000	6.6400	3.3700	3.1100	4.9700	0	3.4125
Hengshui	7.2000	4.6667	7.8100	3.1889	3.4000	3.3500	4.1000	3.9400	3.7333	2.2000	5.7600	3.4625	0

Note: This is the railway passenger flow matrix after data cleaning. “na” represents missed data points. See Appendix A for the complete datasets.

**Table 3 entropy-24-01792-t003:** The global and local statistics for testing and evaluating the dual gravity models.

Statistic		Railway Model		Highway Model	
		Before Data Cleaning	After Data Cleaning	Before Data Cleaning	After Data Cleaning
**Goodness of fit**	*R* ^2^	0.7435	0.7431	0.7900	0.8041
	Adjusted *R*^2^	0.7380	0.7376	0.7851	0.7994
**Standard error (STE)**		0.2531	0.2560	0.2903	0.2881
**Coefficient of variation (CV)**		0.1394	0.1483	0.1958	0.2074
**Number of sample points**		144	144	131	131
** *F* **	Statistic	135.2378	134.9560	159.2751	173.7162
	Sig.	3.5929 × 10^−41^	4.0037 × 10^−41^	7.3138 × 10^−43^	9.1214 × 10^−45^
***p*-value**	ln*K*	4.2035 × 10^−6^	2.3291 × 10^−6^	5.9591 × 10^−8^	2.2355 × 10^−8^
	*u*	2.4816 × 10^−24^	3.3207 × 10^−24^	1.5100 × 10^−8^	4.6746 × 10^−9^
	*v*	1.4308 × 10^−21^	1.9777 × 10^−21^	3.0022 × 10^−7^	3.6536 × 10^−7^
	*σ*	2.9353 × 10^−19^	2.0378 × 10^−19^	3.3192 × 10^−41^	3.3413 × 10^−43^

Note: The Sig. value is the spurious probability value of the F statistic, and the *p*-value is the spurious probability value of the student’s t statistic. If the probability value is less than 0.05, the corresponding statistic can pass the 95% confidence test; if the probability value is less than 0.01, the corresponding statistic can pass the 99% confidence test. The rest can be deduced in this way.

**Table 4 entropy-24-01792-t004:** Partial predicted results for railway passenger flows in the Beijing–Tianjin–Hebei region (average values).

	Beijing	Tianjin	Shijiazhuang	Tangshan	Qinhuangdao	Handan	Xingtai	Baoding	Zhangjiakou	Chengde	Cangzhou	Langfang	Hengshui
Beijing	0	19.1492	10.2015	13.2289	8.0987	7.3460	6.5382	11.0226	10.4151	8.8789	8.3132	16.2535	7.2345
Tianjin	18.9494	0	9.1311	13.7065	7.7131	6.8659	6.0687	9.4598	7.1268	6.9857	10.4808	12.1607	7.1573
Shijiazhuang	9.8257	8.8874	0	5.8317	4.0055	7.0753	7.1020	7.5049	4.9576	3.7760	5.0901	5.1740	6.7488
Tangshan	12.7711	13.3717	5.8453	0	7.4752	4.5374	3.9666	5.6602	5.0783	6.2679	5.4438	7.1379	4.3785
Qinhuangdao	7.6452	7.3581	3.9259	7.3096	0	3.1499	2.7243	3.6137	3.4189	4.4213	3.3853	4.0426	2.9003
Handan	6.9814	6.5941	6.9814	4.4668	3.1711	0	8.4986	4.5778	3.5164	2.8762	3.8086	3.6970	4.8192
Xingtai	6.1198	5.7402	6.9018	3.8459	2.7012	8.3701	0	4.1478	3.0796	2.4757	3.3283	3.2361	4.4406
Baoding	10.4262	9.0423	7.3704	5.5459	3.6209	4.5562	4.1916	0	4.6461	3.5230	4.9283	5.6018	5.0238
Zhangjiakou	9.8378	6.8027	4.8619	4.9688	3.4209	3.4949	3.1077	4.6396	0	3.7653	3.3052	4.4719	3.1907
Chengde	8.3017	6.6004	3.6655	6.0704	4.3790	2.8296	2.4730	3.4824	3.7271	0	2.9744	4.0851	2.6173
Cangzhou	7.7470	9.8701	4.9249	5.2549	3.3419	3.7346	3.3137	4.8554	3.2608	2.9646	0	4.5373	4.3148
Langfang	15.2916	11.5617	5.0540	6.9562	4.0289	3.6598	3.2528	5.5717	4.4542	4.1106	4.5808	0	3.7157
Hengshui	6.7381	6.7365	6.5262	4.2243	2.8615	4.7229	4.4187	4.9468	3.1462	2.6072	4.3125	3.6784	0

Note: This is the predicted railway passenger flow matrix based on data cleaning results. See Appendix A for the complete predicted values.

**Table 5 entropy-24-01792-t005:** Spatial connection strength based on railway passenger flows between the cities in the Beijing–Tianjin–Hebei region (after data cleaning).

	Beijing	Tianjin	Shijiazhuang	Tangshan	Qinhuangdao	Handan	Xingtai	Baoding	Zhangjiakou	Chengde	Cangzhou	Langfang	Hengshui
Beijing	0	0.4221	0.0250	0.0787	0.0087	0.0057	0.0033	0.0337	0.0262	0.0127	0.0094	0.1838	0.0051
Tianjin	0.4221	0	0.0157	0.0941	0.0072	0.0044	0.0024	0.0176	0.0051	0.0045	0.0268	0.0525	0.0050
Shijiazhuang	0.0250	0.0157	0	0.0023	0.0004	0.0053	0.0052	0.0068	0.0011	0.0003	0.0012	0.0013	0.0041
Tangshan	0.0787	0.0941	0.0023	0	0.0066	0.0007	0.0004	0.0019	0.0012	0.0030	0.0016	0.0053	0.0006
Qinhuangdao	0.0087	0.0072	0.0004	0.0066	0	0.0002	0.0001	0.0003	0.0002	0.0007	0.0002	0.0005	0.0001
Handan	0.0057	0.0044	0.0053	0.0007	0.0002	0	0.0118	0.0008	0.0002	0.0001	0.0003	0.0003	0.0010
Xingtai	0.0033	0.0024	0.0052	0.0004	0.0001	0.0118	0	0.0005	0.0001	0.0001	0.0002	0.0002	0.0007
Baoding	0.0337	0.0176	0.0068	0.0019	0.0003	0.0008	0.0005	0	0.0009	0.0002	0.0011	0.0019	0.0012
Zhangjiakou	0.0262	0.0051	0.0011	0.0012	0.0002	0.0002	0.0001	0.0009	0	0.0003	0.0002	0.0007	0.0002
Chengde	0.0127	0.0045	0.0003	0.0030	0.0007	0.0001	0.0001	0.0002	0.0003	0	0.0001	0.0005	0.0001
Cangzhou	0.0094	0.0268	0.0012	0.0016	0.0002	0.0003	0.0002	0.0011	0.0002	0.0001	0	0.0008	0.0006
Langfang	0.1838	0.0525	0.0013	0.0053	0.0005	0.0003	0.0002	0.0019	0.0007	0.0005	0.0008	0	0.0003
Hengshui	0.0051	0.0050	0.0041	0.0006	0.0001	0.0010	0.0007	0.0012	0.0002	0.0001	0.0006	0.0003	0

Note: This is the predicted railway spatial connection strength matrix based on data cleaning results. See Appendix A for the complete predicted values.

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
