# Peer review of "Exploring Spatial Patterns of Interurban Passenger Flows Using Dual Gravity Models"

_entropy, 2022, doi:10.3390/e24121792_

Round 1
Reviewer 1 Report
This paper attempts to build a model of intercity mobility based on Chen’s dual gravity model. They claim that this could solve the missing data problem. However, there are some concerns.
1)Contributions. the author should clarify how the paper applies the dual gravity model. Now it seems that it just makes an estimation based on OLS and generates the missing data. Why does the dual gravity model matter?
2)Methodology: more work should be done on testing how well the missing data is generated. (what is the benchmark?) Also, I don’t understand how fractal methods are related to this work.
3)Data: the authors argue that there are outliers in Spring Festivals and other holidays. However, it seems to me that it is a characteristic of China’s intercity mobility. Data cleaning should not delete this data just based on the R2.
In addition, more control variables should be included, such as industry structure and amenities.
Author Response
Please see the attached file entitled "response to reviewers' comments".

Reviewer 2 Report
In this manuscript, dual gravity models are fitted to characterize the interurban paseenger flows in the Beijing-Tianjing-Hebei region, China, based on the Tencent location data of railway and highway passenger flows. The results show that the railway passenger flows follow the gravity scaling law better than the highway passenger flows and the center of gravity of spatial dynamics has shifted from the Beijing-Tianjing-Tangshan triangle to the Beijing-Baoding-Shijiazhuang axis. In general, this manuscript is well written and the results are convincible. I only have the following minor suggestions that the author need to consider when revising the manuscript.
(1) In the first paragraph of Section 2.1, the citing format of the author’s names for the references [19] and [20] should be consistent with that of the others.
(2) The second mathematical notation “rij” in Line 98 should be “rji”; two minus signs in Table 1 are mistyped as a hyphen; the multiplication sign in Equation (10) and (15) should be typed as “x” insdead of “*”; the values of b in Lines 300 and 302 should be same as those in Equations (10) and (15), respectively.
(3) It seems unnecessary to repeatedly state the specific content of Tobler’s first law of geography that “everything is related to … … than distant things” in the first paragraph of Section 2.1 and in Lines 227 and 228.
(4) I suggest that Table 3 be re-arranged to include the output items as completely as possible. Generally, the outputs of a linear regression model should include two parts. One is the “Variance Analysis” including “SSR”, “SSE”, “SST”, “degrees of freedom”, “F-valus”, “p-value”, etc., indicating whether or not the whole regression equation built is significant. The other is the “Parameter Estimation” including “variable”, “parameter estimator”, “standard error”, “t-value”, “p-value”, etc., showing whether or not the impact of each explanatory variable on the response is significant. The contents in the current table are somewhat confused.
Moreover, in the linear regression analysis, “R-square” is called “coefficient of multiple determination” while “R” is termed as “coefficient of multiple correlation”, which are differen in interpretation. Please clarify that “Multiple R” in Table 3 (also in Table 1) refers to “R” or “R-square”.
(5) Although this manuscript is in gereal well written, there are still several grammar errors. Some examples are as follows:
(i) Line 170: “Because the travel distance … … compared to nationwide” is not a complete sentence; also, should “slightly” in this sentence be “slight”?
(ii) Line 182: it would be better to change “sample numbers” to “sample sizes”;
(iii) Line 318: “the” should be “The”;
(iv) Line 422: “is” should be “are”.
Author Response

(The authors gave the same response as above.)

Round 2
Reviewer 1 Report
The paper has been improved and can be considered for publication. I suggest the authors cite some related articles on the improvement of the gravity model in recent years and compare your gravity models with these past models, and future clarify your advantages and deficiencies.
Gu, H., Shen, J., & Chu, J. (2022). Understanding Intercity Mobility Patterns in Rapidly Urbanizing China, 2015–2019: Evidence from Longitudinal Poisson Gravity Modeling. Annals of the American Association of Geographers. https://doi.org/10.1080/24694452.2022.2097050
Shen, J. (2016). Error analysis of regional migration modeling. Annals of the American Association of Geographers, 106(6), 1253-1267.
Gu, H., & Shen, T. (2021). Modelling skilled and less‐skilled internal migrations in China, 2010–2015: Application of an eigenvector spatial filtering hurdle gravity approach. Population, Space and Place, 27(6), e2439.
Author Response
Dear reviewer, Thank you very much for providing some latest literature about the topic. We’ve improved the quality of our manuscript by your suggested references to the bibliographic list. Besides, some format details are modified. Our improvements are listed as follows. [Improvement1] We have introduced the literature and compared our models with them to further clarify the advantages and disadvantages as follows. “There’re some newly developed gravity models with satisfactory data fitting and predicting abilities in recent years, such as the Poisson pseudo-maximum likelihood estimation gravity model, the eigenvector spatial filtering hurdle gravity model, and the network spatially filtered Poisson migration model [59-61]. These models consider the spatial autocorrelation problem and introduce some control variables. In practice, these models can be used for urban governance because they can tell which determinants affect migrations more. Our models can be used for macro regional planning because the changing regional development focus trend can be detected and the transportation network fractal structure can be characterized with our models. Compared with these newly developed models, our models have both advantages and deficiencies. The advantages of our study lie in three aspects……” [Improvement2] We have standardized the format of tables and figures. Some numbers typed wrong in Table 1, Equation 11, and Equation 16 are modified. [Improvement3] We have deleted some repetitive sentences in Line 110, Line 116. [Improvement4] We have corrected some grammar errors, such as “penetrate rate” in Line 171 should be “penetration rate”. [Improvement5] We have unified “File S1” to “File 1” to be consistent with our supplement files.
Reviewer 2 Report
The authors well addressed the issues that I raised in the first round review. I therefore recommend that the manuscript be accepted in present form.
Author Response
Thank you very much.